# Nitrate and Nitrite Metabolism in Aging Rats: A Comparative Study

**DOI:** 10.3390/nu15112490

**Published:** 2023-05-26

**Authors:** Barbora Piknova, Ji Won Park, Samantha M. Thomas, Khalid J. Tunau-Spencer, Alan N. Schechter

**Affiliations:** Molecular Medicine Branch, National Institute of Diabetes and Digestive and Kidney Diseases (NIDDK), National Institutes of Health, Bethesda, MD 20892, USA

**Keywords:** nitric oxide, nitrate, nitrite, aging

## Abstract

Nitric oxide (NO) (co)regulates many physiological processes in the body. Its short-lived free radicals force synthesis in situ and on-demand, without storage possibility. Local oxygen availability determines the origin of NO—either by synthesis by nitric oxide synthases (NOS) or by the reduction of nitrate to nitrite to NO by nitrate/nitrite reductases. The existence of nitrate reservoirs, mainly in skeletal muscle, assures the local and systemic availability of NO. Aging is accompanied by changes in metabolic pathways, leading to a decrease in NO availability. We explored age-related changes in various rat organs and tissues. We found differences in nitrate and nitrite contents in tissues of old and young rats at baseline levels, with nitrate levels being generally higher and nitrite levels being generally lower in old rats. However, there were no differences in the levels of nitrate-transporting proteins and nitrate reductase between old and young rats, with the exception of in the eye. Increased dietary nitrate led to significantly higher nitrate enrichment in the majority of old rat organs compared to young rats, suggesting that the nitrate reduction pathway is not affected by aging. We hypothesize that age-related NO accessibility changes originate either from the NOS pathway or from changes in NO downstream signaling (sGC/PDE5). Both possibilities need further investigation.

## 1. Introduction

Nitric oxide is one of the most versatile signaling molecules in nature. Its short-lived nature and reactivity with other free radicals predetermine the needs of its synthesis in situ, without the possibility of storage. The origin of NO in the mammalian body depends on oxygen availability in tissues. NO can either be directly synthetized by a family of nitric oxide synthases (NOS) [1] or created during the two-step reduction of nitrate to nitrite by molybdenum-containing proteins (xanthine oxidoreductase (XOR), aldehyde oxidase (AO), sulfite oxidase (SO) and mitochondrial amidoxime-reducing component (mARC)) [2] and then the reduction of nitrite to NO by molybdenum- or heme-containing proteins [3]. The second pathway is predominant in a low-oxygen environment, such as that found in the tissues of most internal organs [4]. Recently, we showed that some organs, such as skeletal muscle, contain higher levels of nitrate, and can be regarded as a global body nitrate reservoir [5,6]. Other organs, such as the liver, show higher XOR activity/levels and can be regarded as sites of nitrate reduction to nitrite [7]. After nitrate reduction by XOR, nitrite is then transported by the bloodstream and reduced to NO on the site of demand by deoxy heme-proteins [8,9,10,11] or further reduced to NO by XOR or AO tissue [12,13,14,15,16,17,18]. On the cellular level, nitrate/nitrite transporters, such as sialin and CLC-proteins, shuffle these ions into and from cells [19,20,21,22]. To complete the picture, nitrate originates from several sources, such as diet, the oxidation of nitrite and unused NO by oxyhemoglobin and oxymyoglobin, and the direct synthesis of nitrate by NOS, mainly NOS1, during its so-called futile cycle [23,24,25]. 

Here, we investigate how this delicate balance of the storage, transport and reduction of nitrate/nitrite ions is affected/disturbed by processes naturally occurring during aging. We concentrate mainly on changes in the skeletal muscle, cardiovascular system, brain, and eye as these organs and systems (in humans at least) are likely to be the most affected by age-related conditions, such as sarcopenia, cardiovascular and neurodegenerative/cognitive diseases and/or glaucoma or macular degeneration.

We studied these age-related changes at the baseline state and after dietary intervention where animals consumed a high nitrate diet for 5 days. The main purpose of this approach was to disturb the existing baseline equilibrium to determine whether nitrate/nitrite physiology and metabolism are affected by age. Secondly, dietary nitrate supplementation is used quite frequently in the exercise field in order to improve endurance [26,27]. It has been known to decrease elevated blood pressure [28] and there is some evidence that it could be beneficial in improving several disease states, such as PAD [29] and Becker muscular dystrophy [30]. All these facts are highly relevant to the aging human population.

## 2. Materials and Methods

Two sets of healthy Wistar rats (Envigo, Indianapolis, IN, USA) aged 3 (“young”, n = 16, 8 males/8 females, weight 240 g) and 18 (“old”, n = 8, 4 males/4 females, weight 350–450 g) months, respectively, were randomly divided into 2 groups: a control group eating a regular rat diet (NIH-07, Envigo, Indianapolis, IN, USA) and a nitrate-supplemented group with 1 g/l of NaNO_3_ (Sigma, St. Louis, MO, USA) added to their drinking water. We chose the dose and duration of nitrate supplementation based on our previous studies where such treatment led to significantly elevated nitrate/nitrite levels in rat organs [6,31,32,33]. The size of the rat groups was determined based on previous reports [34,35]. After 5 days of dietary intervention, rats were euthanized, samples from organs and blood were collected, and nitrate and nitrite levels in plasma and tissues were measured using chemiluminescence, as described in detail in our previously published work [5,36]. The results are shown as mean ± SD. Statistical analysis was performed using ANOVA, *p* < 0.05.

Details regarding all materials, data and protocols associated with the publication are available upon written request. All animal procedures were approved by the NIDDK Animal Care and Use Committee (protocol number K049-MMB-20) and carried out according to recommendations in the Guide for the Care and Use of Laboratory Animals of NIH. This animal study is reported in accordance with ARRIVE guidelines.

## 3. Results

*Note:* We organized each of the two panels in Figure 1, Figure 2 and Figure 3 into three groups. The first group—liver, plasma and gluteus—represent events at muscle and whole-body levels; the second group—eye, lacrimal glands and brain—mirror events at the neuronal level; and the third group—salivary glands, heart, aorta and vena cava, together with plasma—reflect the cardiovascular system.

### 3.1. Nitrate and Nitrite in Young Rats at Baseline vs. after Nitrate Treatment

Nitrate and nitrite levels at baseline and after 5 days of nitrate treatment for young rats are plotted in Figure 1. As expected, dietary nitrate supplementation led to, in most cases, a significant increase in nitrate and nitrite levels in the studied organs and tissues.

At baseline in young rats (Figure 1A, light gray bars), we observed a nitrate gradient from gluteus to plasma to liver, with nitrate content per gram tissue highest in the gluteus (34.2 ± 16.5 nmol/g), intermediate in plasma (15.1 ± 3.9 nmol/g) and lowest in the liver (11.3 ± 3.5 nmol/g). The values for nitrate in several other muscles (tibias anterior (TA), extensor digitorum longus (EDL), soleus, gastrocnemius) are summarized in Appendix A. Baseline nitrate concentrations in the eyes, lacrimal glands and brains of young rats were 8.5 ± 0.4; 15.5 ± 4.5 and 16.4 ± 5.9 nmol/g, respectively. Nitrate concentrations in the salivary glands, heart, aorta and vena cava were 8.4 ± 3.5, 22.9 ± 3.6, 10.5 ± 2.3 and 20.9 ± 7.2 nmol/g, respectively.

After dietary nitrate treatment, nitrate contents in young rats increased several folds in all organs and tissues when compared to baseline values (Figure 1A, dark gray bars). Plasma, as a nitrate carrier, increased to 58.8 ± 30.8 nmol/g, gluteus increased to 48.8 ± 10.4 nmol/g and liver increased to 15.7 ± 6.3 nmol/g. The values of nitrate concentration in several other muscles after supplementation (tibias anterior (TA), extensor digitorum longus (EDL), soleus, gastrocnemius) are summarized in Appendix A. Nitrate concentrations were 18.9 ± 7.4 nmol/g in the brain, 15.5 ± 2.0 nmol/g in the eye and 26.2 ± 9.1 nmol/g in the lacrimal glands. The nitrate concentrations across cardiovascular organs were elevated compared to baseline values to similar values across all organs, with 27.5 ± 5.9 nmol/g in the vena cava, 26.0 ± 10.1 nmol/g in the heart, 24.6 ± 10.2 in the aorta and 20.1 ± 9.4 nmol/g in the salivary glands.

The relative enrichment of all organs by nitrate caused by dietary nitrate supplementation is summarized in Appendix A. As expected, the greatest nitrate increase was observed in plasma (3.9-fold), followed by the salivary glands (2.4-fold), aorta (2.3-fold), eye (1.8-fold), lacrimal glands (1.7-fold), muscle (1.4-fold), liver (1.4-fold), vena cava (1.3-fold), brain (1.2-fold) and heart (1.1-fold).

In general, nitrite at baseline in the tissues of young rats (Figure 1B, light gray bars) was much more uniformly distributed than nitrate. Nitrite concentrations in the liver and gluteus were similar (0.36 ± 0.11 and 0.40 ± 0.14 nmol/g, respectively), which was in the same range as values for other muscle groups (TA, EDL, soleus, gastrocnemius; see Appendix A). Plasma contained 0.24 ± 0.08 nmol/g of nitrite, which was the lowest concentration from all studied organs and tissues. Nitrite concentration in the brain was 0.82 ± 0.38 nmol/g, which was the third highest of all the organs studied, with the lacrimal glands and eye containing 0.62 ± 0.11 and 0.25 ± 0.01 nmol/g, respectively. From the organs of the cardiovascular system, the vena cava was the tissue with the highest concentration of nitrite (1.65 ± 0.42 nmol/g), followed by the aorta (1.02 ± 0.43 nmol/g). The heart and salivary glands contained 0.32 ± 0.09 and 0.32 ± 0.13 nmol/g of nitrite, respectively.

After dietary nitrate treatment, nitrite contents in young rats increased several folds in all organs and tissues when compared to baseline values (Figure 1B, dark gray bars). Nitrite content increased to 0.59 ± 0.15 nmol/g in plasma, 0.76 ± 0.20 nmol/g in the gluteus and 0.44 ± 0.14 nmol/g in the liver. The values for nitrate in several other muscles (tibias anterior (TA), extensor digitorum longus (EDL), soleus, gastrocnemius) are summarized in Appendix A. The concentration of nitrite increased to 0.92 ± 0.14 nmol/g in the brain, 0.78 ± 0.29 nmol/g in the lacrimal glands and 0.49 ± 0.11 nmol/g in the eye. The circulatory system organs reached levels of 2.31 ± 0.86 nmol/g in the aorta, 1.79 ± 0.22 nmol/g in the vena cava, 0.43 ± 0.19 nmol/g in the salivary glands and 0.41 ± 0.10 in the heart. The relative enrichment of all organs by nitrite caused by dietary nitrate supplementation is summarized in Appendix A. The greatest increase was observed in plasma (2.5-fold), followed by the aorta (2.3-fold), eye (2-fold), gluteus (1.9-fold), salivary glands (1.3-fold), heart (1.3-fold), lacrimal glands (1.3-fold), liver (1.2-fold), brain (1.1-fold) and vena cava (1.1-fold).

### 3.2. Nitrate and Nitrite in Old Rats at Baseline vs. after Nitrate Treatment

Nitrate and nitrite levels measured at baseline and after 5 days of nitrate treatment for old rats are plotted in Figure 2. As expected, and similar to young rats, dietary nitrate supplementation led to, in most cases, a significant increase in nitrate and nitrite detected in the studied organs and tissues of old rats.

At baseline (Figure 2A, light green bars), liver, plasma and gluteus nitrate levels reached 12.5 ± 7.2, 7.6 ± 1.6 and 50.0 ± 16.6 nmol/g tissue, respectively. The nitrate concentrations for other muscle groups studied (TA, EDL, soleus, gastrocnemius) are shown in Appendix A. Nitrate concentrations in the eyes, lacrimal glands and brains of old rats were 1.04 ± 0.1, 11.1 ± 5.3 and 26.9 ± 4.5 nmol/g, respectively. Nitrate concentrations in the salivary glands, heart, aorta and vena cava reached 10.3 ± 6.6, 14.7 ± 6.0, 8.9 ± 1.5 and 13.2 ± 7.2 nmol/g, respectively. After 5 days of nitrate supplementation, there was a major increase in nitrate concentrations in all measured organs (Figure 2A, dark green bars). Nitrate concentrations increased to 65.8 ± 33.9 nmol/g in plasma, as a major dietary nitrate transporter; 57.1 ± 24.4 nmol/g in the gluteus; and 27.4 ± 10.5 nmol/g in the liver. The nitrate concentrations in other muscle groups (TA, EDL, soleus, gastrocnemius) are shown in Appendix A. Nitrate levels increased to 62.6 ± 15.5 nmol/g in the lacrimal glands, 29.3 ± 4.5 nmol/g in the brain to and 3.3 ± 1.6 nmol/g in the eye. The circulatory system organs reached values of 45.0 ± 20.8 nmol/g in salivary glands, 34.9 ± 10.7 in the aorta, 23.3 ± 9.4 in the heart and 21.2 ± 8.8 nmol/g in the vena cava. The relative enrichment of all organs by nitrate caused by dietary nitrate supplementation is summarized in Appendix A and plotted in Appendix A. Five days of dietary nitrate treatment led to a major increase in nitrate contents in all organs and tissues of old rats, with the greatest increase observed in plasma (8.7-fold), followed by the lacrimal glands (5.6-fold), salivary glands (4.4-fold), aorta (3.9-fold), eye (3.3-fold), vena cava (1.6-fold), heart (1.6-fold), gluteus (1.1-fold) and brain (1.1-fold).

Baseline nitrite contents in the organs of old rats (Figure 2B, light green bars) were in a similar range to those of young rats. Nitrite concentrations in the liver, plasma and gluteus were uniform, with 0.24 ± 0.07, 0.27 ± 0.03 and 0.25 ± 0.07 nmol/g, respectively. The nitrite values for other muscle groups (TA, EDL, soleus, gastrocnemius) are in Appendix A. Similar to young rats, the brains of old rats contained the second-highest nitrite concentration (0.96 ± 0.22 nmol/g). Nitrite in the lacrimal glands reached 0.31 ± 0.18 nmol/g, and nitrite concentrations in the eye were the lowest of all studied organs, at only 0.18 ± 0.06 nmol/g. From the organs of the cardiovascular system, the vena cava was the tissue with the second highest concentration of nitrite (0.77 ± 0.24 nmol/g), followed by the aorta (0.53 ± 0.11 nmol/g), heart (0.37 ± 0.05 nmol/g), plasma (0.27 ± 0.03 nmol/g) and salivary glands (0.20 ± 0.05 nmol/g).

After nitrate supplementation, nitrite values increased in all organs (Figure 2B, dark green bars). The nitrate concentration was 0.46 ± 0.08 nmol/g in plasma, 0.40 ± 0.11nmol/g in the liver and 0.30 ± 0.04 nmol/g in the gluteus (see Appendix A for values in TA, EDL, soleus and gastrocnemius). The largest increase in nitrite concentration in old rats was noted in the brain (1.13 ± 0.54 nmol/g), while that in the lacrimal glands and eye reached 0.37 ± 0.34 and 0.24 ± 0.02 nmol/g, respectively. On the circulatory side, there was a large increase in nitrate concentration in the aorta (1.48 ± 0.58 nmol/g), followed by the vena cava (0.99 ± 0.82 nmol/g), heart (0.69 ± 0.20 nmol/g) and salivary glands (0.26 ± 0.08 nmol/g).

After dietary nitrate treatment, nitrite contents in old rats increased in all organs and tissues compared to baseline values (Appendix A). The greatest increase was observed in the aorta (2.8-fold), followed by the heart (1.9-fold), plasma (1.7-fold), liver (1.7-fold), eye (1.3-fold), salivary glands (1.3-fold), vena cava (1.3-fold), gluteus (1.2-fold), lacrimal glands (1.2-fold) and brain (1.2-fold). In skeletal muscles, nitrite increased only modestly, by 1.2-fold in the gluteus, with similar modest increases in all other measured skeletal muscles (TA, EDL, soleus, gastrocnemius).

### 3.3. Nitrite Levels in Young vs. Old Rats at Baseline and after Nitrate Treatment

We calculated the ratios of nitrate and nitrite concentrations related to age at baseline and after 5 days of dietary nitrate supplementation (Figure 3) to compare the effects of nitrate and nitrite in these groups. Appendix A present the absolute values of nitrate and nitrite at baseline and after nitrate supplementation that were already described in previous sections for young and old rats. The calculated ratios are summarized in Appendix A.

The distribution of nitrate in the bodies of old rats differed from that in young rats at baseline levels (Figure 3A, light orange bars). Old rats had higher nitrate concentrations in the brain (1.6-fold), gluteus (1.5-fold) and liver (1.1-fold) compared to young rats. In all other organs, nitrate concentrations were lower in old rats than in young rats in the aorta (0.9-fold), lacrimal glands (0.7-fold), heart (0.6-fold), vena cava (0.6-fold) and plasma (0.5-fold), with the most significant difference observed in the eye (only 0.1-fold).

After nitrate supplementation (Figure 3A, dark orange bars), differences were attenuated, but some persisted. Nitrate concentrations in salivary and lacrimal glands in old rats became higher than those in young rats (2.4- and 2.2-fold, respectively) and in the liver, the difference reached 1.8-fold. The brains and aortas of old rats had 1.6- and 1.4-fold higher nitrate contents, respectively, than those of young rats. The gluteus had a 1.2-fold enrichment and plasma had a 1.1-fold enrichment. The heart, vena cava and, in particular, the eye, had lower nitrate contents than their young counterparts, with 0.9-, 0.8- and 0.21-fold, respectively.

At baseline levels, there were substantial differences in nitrite distributions among the organs of young rats compared to old rats (Figure 3B, light orange bars). Old rats had higher contents of nitrite in the plasma (1.1-fold), brain (1.2-fold) and heart (1.2-fold). Nitrite concentrations were lower in the organs of old rats than in those of young rats for all other organs, including the eye (0.7-fold), liver (0.7-fold), gluteus (0.6-fold), salivary glands (0.6-fold), aorta (0.5-fold), lacrimal glands (0.5-fold) and vena cava (0.5-fold).

After nitrate supplementation, the differences persisted, but were attenuated (Figure 3B, dark orange bars). Old rats still had higher nitrate contents in the heart (1.7-fold) and brain (1.2-fold), but a comparable amount of nitrate in the liver (0.9-fold). Nitrite concentrations were lower in old rats in all other organs and tissues, including the plasma (0.8-fold), aorta (0.6-fold), vena cava (0.6-fold), salivary glands (0.6-fold), eye (0.5-fold), lacrimal glands (0.5-fold) and gluteus (0.4-fold).

### 3.4. Protein Level Changes in Young and Old Rats at Baseline and after Nitrate Treatment

We evaluated the levels of proteins involved in nitrate transport (sialin) and the reduction of nitrate to nitrite and nitrite to NO (xanthine oxidoreductase, XOR) in the liver, gluteus and eye using Western blotting (Figure 4). Except for the eye, there was no difference in sialin or XOR expression between old and young rats. After nitrate supplementation, there was no significant change in the levels of sialin or XOR compared to baseline levels.

## 4. Discussion

Over the past decade, there has been a noticeable shift of focus away from NOS enzymes as an almost exclusive source of NO towards the nitrate/nitrite reductive pathway as an equally important NO generator. Being less reactive than NO, both ions serve as an almost-immediate source of NO and nitrate in particular can be stored in the body and used when the need arises. Based on our previous work with rodents, we showed that nitrate concentrations in skeletal muscle (in particular, in the gluteus) exceed that observed in the bloodstream and other tissues several-fold [5]. We formulated a hypothesis postulating that the skeletal muscle tissue as a mammalian body nitrate reservoir is freely accessible to the bloodstream to be transported and used in sites of NO deficiency and, similar to the liver [7], the skeletal muscle itself can generate nitrite/NO by the reduction of nitrate to nitrite and NO, using XOR as a nitrate reductase [37]. We also showed that this reservoir reacts to dietary nitrate changes by either increasing its storage (high nitrate diet) or releasing more nitrate into the bloodstream for use (dietary nitrate deficiency) [31,33]. We confirmed a similar dynamic in humans [27,38,39]. Here, we discuss the effect of age on the levels/availability of nitric oxide metabolites in healthy, aged rats.

### 4.1. Baseline Conditions

#### 4.1.1. Baseline Nitrate

Consistent with our previous reports, nitrate distribution among the organs and tissues of young rats was nonuniform, with nitrate concentration gradients present from the gluteus through plasma and liver, with gluteus nitrate concentrations 3-fold higher than those in the liver and plasma nitrate 1.4-fold higher than that in the liver. This is consistent with the idea of skeletal muscle serving as a nitrate reservoir, a hypothesis we previously formulated and investigated [5,31,33,37]. However, this picture significantly differs for the group of old rats at baseline. Here, the gluteus still seems to act as a nitrate reservoir, with a 4-fold excess of nitrate over the liver, with other collected muscles (TA, EDL, soleus, gastrocnemius) supporting this finding (6.5-, 5.4-, 5.4- and 3.2-fold, respectively). However, nitrate levels in plasma are significantly lower than those in the liver or gluteus (0.6- and 0.2-fold, respectively). Interestingly, when compared to young rats, nitrate levels in the gluteus and brains of old rats were greater by 1.5- and 1.6-fold, respectively, but the blood of old rats contained only half the nitrate levels (0.5-fold) of young rats. Other significant changes included those observed in the eye and vena cava (0.1-fold and 0.6-fold changes in old rats vs. young ones).

A first look at the picture reveals that, in old rats, the function of skeletal muscle as a place for nitrate storage, and likely that of the liver as a nitrate reduction “factory,” seems to be intact, as the nitrate concentration gradient from muscle to liver persists, implying the existence of active nitrate transport between tissues and blood. However, it seems that the ability of blood to act as a transporting medium in old rats is significantly diminished for reasons that are not yet well understood. There are no differences in the expression of the nitrate transporter (sialin) or nitrate reductase (XOR) proteins in the skeletal muscle (or liver), as revealed by the Western blot results; thus, the levels of transporting and reducing proteins are likely unaffected by age in these tissues. Currently, we do not have information about their activity levels, which could be playing a significant role. We can only speculate on what could prevent the diffusion of stored nitrate from muscle into the bloodstream, for example, age-related changes in the vessel walls themselves [40]. Another possibility is a lower supply of endogenously generated NO and nitrate by endothelium and other organs due to age-related changes in NOS expression/activity in general and, in particular, in vessel walls [41]. Diminishing NOS activity in blood vessels would result in lower NO (and nitrate) levels in the blood since NOS is one of the major endogenous nitrate sources via its futile cycle [24]. One factor in favor of such changes might be that all other parts of the circulatory system of old rats (heart, aorta, vena cava) showed slightly diminished nitrate concentrations. One should also keep in mind the possibility of increased oxidative stress due to NOS uncoupling and increased XO activity in the vascular system [42,43,44], which may add to changes in NO production on both sides of the pathway (NOS- and nitrate-related). As this study is the first attempt to establish the array of changes in nitrate metabolism related to aging, unfortunately, we can only offer the aforementioned “educated guesses,” as more detailed work related to the NOS part of the NO equation must be conducted in order to answer such questions.

Possibly one of the most surprising findings here is that the brains of old rats are significantly enriched by nitrate and their eyes are dramatically depleted of nitrate compared to their young counterparts. The brain and eye both seem to be very good candidates for the use of the nitrate reduction pathway to support function [45,46,47]. The eye is also the only place with significantly altered expression of sialin and XOR: expression of both was 2.6- and 2-fold greater, respectively, in old rats compared to young rats, which is not that surprising when thinking about the importance of NO for eye function [48]. More research must be conducted to understand the actions and functional significance of NO and the consequences of a lack of NO in the eye.

#### 4.1.2. Baseline Nitrite

Nitrite was, in general, almost uniformly distributed among the plasma, eyes, livers, salivary glands and hearts of young rats. This picture is consistent with the notion of the “local” use of nitrate by organs—nitrate is reduced by residing nitrate reductase (XOR and possibly other MoCo-proteins) into nitrite (and then to NO) when and where the need for NO arises. We also noticed that nitrite is elevated in the brains and hearts of old rats compared to young ones. The brain is the most energy-demanding organ in mammals [49], with extensive blood flow that is very tightly regulated to ensure a constant supply of oxygen and nutrients [50]. The NO/endothelin system plays a crucial role in balancing the vasodilation/vasoconstriction of blood vessels and ensuring constant cerebral blood flow. As we showed in our previous work, nitrite is present in cerebral circulation and is one of the sources of NO that can compensate for impaired NOS function [46]. Therefore, one can easily expect and explain a 3.4-fold increase in nitrite concentrations in this tissue when compared to plasma. A similar explanation could be true for nitrite increases in the aorta and vena cava (4.3- and 6.9-fold increases over plasma, respectively). As two major blood vessels, the aorta and vena cava diameters and the associated blood pressure within them must also be under tight control to assure the proper functioning of all organs. However, this hypothesis requires experimental evidence.

Nitrite in old rats follows the same distribution pattern as that described for young rats. Nitrite levels in old rat organs tended, in general, to be lower than those in young rats, except for slightly elevated nitrite levels in the plasma, brain and heart (1.1-, 1.2- and 1.2-fold increases, respectively). This would suggest that XOR reduces nitrate to nitrite with similar (or better) efficiency in old and young animals, as nitrate levels in old rats are, in general, higher than those in young rats. However, because the XOR active site (MoCo cofactor) is used for nitrate- and nitrite-reductase activity, it is possible that the higher nitrate contents in old rats might partially inhibit XOR nitrite reduction in NO, as reported in rat vessels by [51]. NO deficiency related to older age could also result from increased oxidative stress leading to a dysfunctional NOS system [52]. Downstream signaling, namely, soluble guanyl cyclase (sGC), or other members of the cascade (such as PDE5) could also malfunction in older age [53,54,55,56,57,58]. We consider that this might be the most likely explanation for our findings, but this hypothesis requires further experimental evidence to be confirmed.

### 4.2. Changes following Dietary Nitrate Supplementation

#### 4.2.1. Nitrate after Dietary Supplementation

When fed a high nitrate diet, nitrate levels increased in all organs of young rats, as expected based on previously reported results. The most significant nitrate increase was observed in plasma (a 3.9-fold increase over baseline values), with other significant increases observed in the salivary glands, aorta, eye and lacrimal glands (2.4-, 2.3-, 1.8 and 1.7-fold, respectively). Nitrate levels also increased in the gluteus (1.4-fold), as well as in all other examined muscles (TA, EDL, soleus, gastrocnemius; 2.1-, 2-, 1.6- and 1.6-fold, respectively). Similarly, in old rats, increased access to dietary nitrate led to a significant increase in the nitrate concentrations of all organs except the brain (which showed only a slight increase over baseline values), with the most notable changes in the plasma (8.7-fold), lacrimal glands (5.7-fold), salivary glands (4.4-fold) and aorta (3.9-fold). Nitrate levels in the gluteus increased slightly (1.1-fold), with similar increases observed in other muscles (TA, soleus, gastrocnemius; 1.6-, 1.1- and 1.1-fold, respectively). This suggests that nitrate transport into and from plasma to cells (sialin, CLC-family of proteins [59]) is well functioning even in old age. Interestingly, in contrast to humans, neither young nor old rat salivary glands showed marked increases in nitrate or nitrite. This could be related to a low contribution of the rat oral microbiome to the nitrate–nitrite–NO reductive pathway [60]. When compared across age groups, nitrate levels in old rats after supplementation were higher than those in young rats, with the notable exception of the eye. This was related to the generally higher nitrate baseline values observed in old rats. Based on the above results, we believe that the organs of old rats are just as able as those of young rats to transport, store and use nitrate. We also believe that differences in NO availability related to aging are likely due either to NOS-related NO generation or to NO downstream signaling.

#### 4.2.2. Nitrite after Dietary Supplementation

Increased access to nitrate led to increased nitrite concentrations in all organs of young and old rats. Similar to the results measured at baseline, the resulting concentrations of nitrite after a high nitrate diet were lower in old rats than in young rats, with the exception of the brain and heart, where nitrite in old rat organs exceeded 1.2-fold and 1.7-fold, respectively, that in young rats. In general, there could be at least two underlying explanations: that less nitrate is reduced to nitrite by XOR or more nitrite is oxidized back to nitrate if conditions are more pro-oxidative. In either case, this would lead to (apparent) nitrite deficiency. Another possibility is that the nitrate reductase activity of XOR is decreased in old age. All the above hypotheses must be explored experimentally.

#### 4.2.3. Summary and Hypothesis

Our findings and hypotheses are summarized in Figure 5. Based on our data, nitrate seems to accumulate in most of the organs of old rats, either from dietary nitrate or by increased NOS futile cycle activity. Despite higher nitrate concentrations in old rats, nitrite concentrations in the majority of their organs were relatively lower when compared with young rats. We believe that a major source of nitrite for rat organs is the nitrate reductase activity of XOR and other MoCo proteins since the oral microbiome is not a significant contributor to nitrite levels in rats [55]. There are several possible causes that could contribute to the observed nitrite deficiency or depletion in old rats. Lower XOR nitrite reductase activity would lead to nitrite deficiency [46], while increased nitrite reduction to NO by heme proteins and/or increased nitrite oxidation back to nitrate (in sites with higher oxidative stress in old rats) would cause nitrite depletion. At this point, we can only speculate about the reason; more experiments are needed to elucidate the underlying processes.

NOS enzymes are also equally important contributors to NO balance and evaluations of the relative contributions of NOS vs. dietary nitrate to NO balance in old vs. young rats are needed. The results could generate new hypotheses about general changes in the NO pathway (including both sources—NOS and nitrate) due to the aging process.

Equally importantly, one could also hypothesize that additional changes in downstream signaling require consideration, especially at the level of soluble guanyl cyclase (sGC), which is susceptible to inactivation due to its heme oxidation and/or PDE5, which destroys cGMP [56].

## 5. Conclusions

Aging is a natural process that brings its own irreversible challenges and manifests itself by changes occurring in all metabolic pathways. One of the changes seems to be decreased access to NO. Previously, we studied the reductive pathways of NO production that use nitrate as a source of NO. We found that, in young, healthy rats, nitrate is sequestered from the bloodstream and stored mainly in skeletal muscle tissue. When there is a dietary shortage of nitrate (or an increased local NO need by some metabolic pathway), the muscle can release the stored nitrate into the bloodstream, transport it to the site of need and use it to fuel its reduction to nitrite and NO. The muscle itself uses the stored nitrate to supply/supplement NO for its functioning, especially during exercise. In the present study, we investigated how this delicate balance of the storage, transport and reduction of nitrate/nitrite ions is affected/disturbed by processes naturally occurring during aging. We concentrated mainly on changes in the skeletal muscle, cardiovascular system, brain and eye as these organs and systems (in humans at least) are likely most affected by age-related conditions, such as sarcopenia, cardiovascular and neurodegenerative/cognitive diseases and/or glaucoma and macular degeneration.

We found that aging does not profoundly affect all of the processes related to the uptake and storage of nitrate as nitrate in the organs of old rats increased in similar patterns to those observed in young rats. The main difference at baseline conditions was the nitrate and nitrite concentrations in the eye, which were significantly lower in old rats than in their young counterparts. Unexpectedly, the decrease of both ions in the eyes of old rats was accompanied by significantly higher expression of the nitrate transporter, sialin, and nitrate reductase, XOR. Interestingly, worsening eyesight and some ocular diseases (glaucoma, macular degeneration) in humans are one of the first changes related to aging. We are currently working on elucidating the role of the nitrate/nitrite pathway in the eye. Additionally, currently, there is little doubt that, with aging, the mammalian body experiences a deficit of NO that might eventually lead to an array of metabolic diseases and syndromes. However, the conclusion from our present work is that as we age, the main changes in NO production are not related directly to the NO reductive pathway but might be caused either by irreversible changes in NOS-related NO production (so-called “endothelium dysfunction”) or by changes in downstream NO signaling—especially by the oxidation of sGC (or increase in the fraction of apo-sGC) or the increased activity of PDE5 [61], as schematically expressed in Figure 5. The first of these hypotheses (aging does not alter the nitrate reductive pathway) leaves open the possibility of using dietary nitrate supplementation to compensate for changes in NOS-related NO production. If, after NO levels increase, there is still an apparent NO deficiency, we believe that this could be a sign of dysregulated downstream signaling, which is a condition requiring a different approach, depending on the exact site of malfunctioning. One could then start considering the use of PDE5 inhibitors or sGC activators. However, currently, there is not enough experimental evidence in favor of either of these hypotheses.

## Figures and Tables

**Figure 1 nutrients-15-02490-f001:**
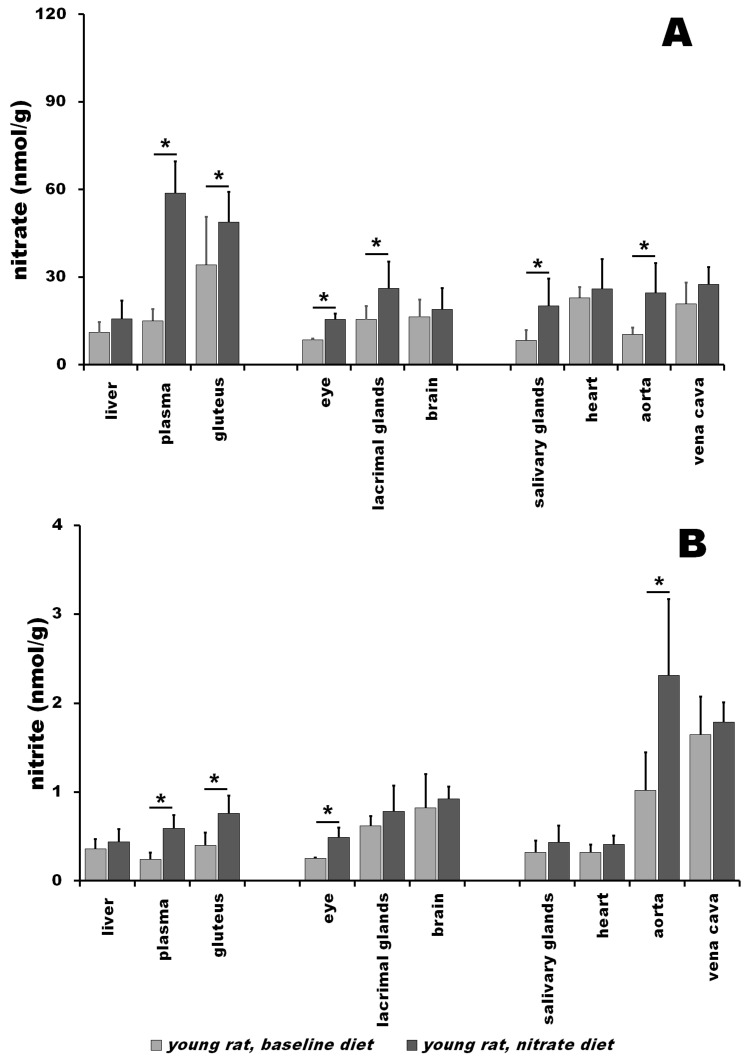
Nitrate (**A**) and nitrite (**B**) levels in young rats at baseline (light gray bar) and after nitrate supplementation for 5 days (dark gray bar). Values are presented as average ± SD, n = 4 for all organs, ***** denotes *p* < 0.05.

**Figure 2 nutrients-15-02490-f002:**
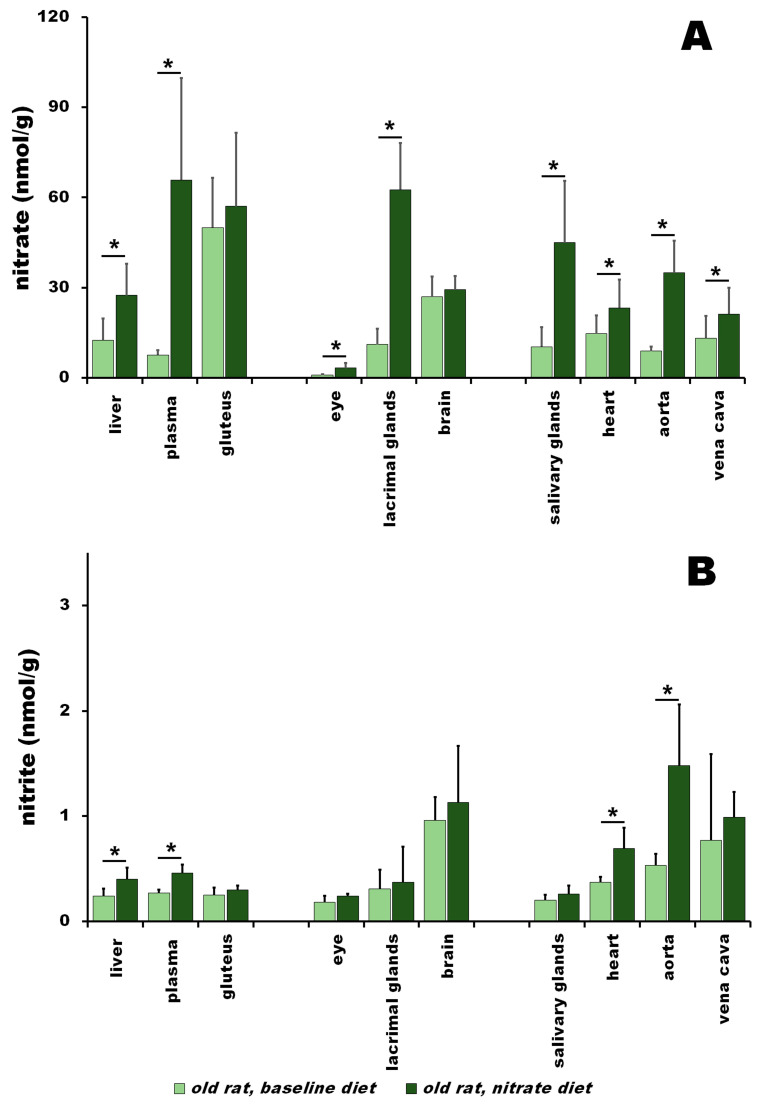
Nitrate (**A**) and nitrite (**B**) levels in old rats at baseline (light green bar) and after nitrate supplementation for 5 days (dark green bar). Values are presented as average ± SD, n = 4 for all organs, ***** denotes *p* < 0.05.

**Figure 3 nutrients-15-02490-f003:**
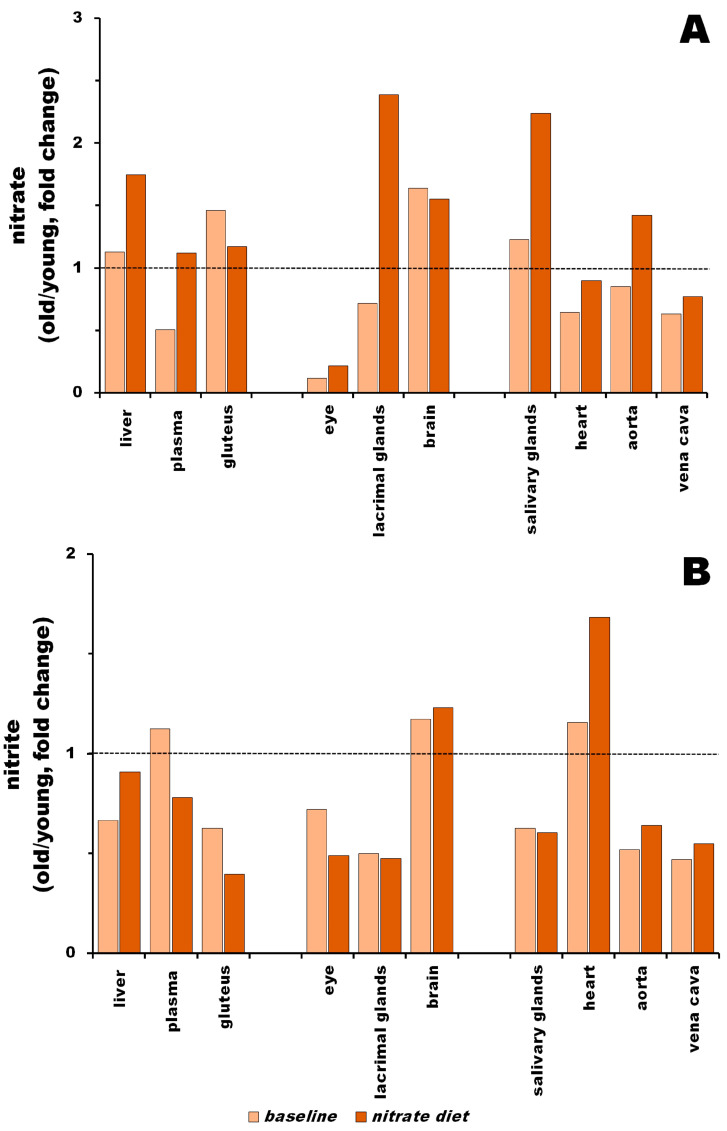
Ratios of nitrate (**A**) and nitrite (**B**) contents in organs and tissues of old vs. young rats at baseline (light orange bars) and after 5 days of nitrate supplementation (dark orange bars). Ratios were calculated as concentrations of nitrate (nitrite) in old rat organs/nitrate (nitrite) in young rat organs at both dietary conditions. Ratio < 1 reflects higher content of nitrate (nitrite) in young rat tissues; ratio > 1 reflects higher content of nitrate (nitrite) in old rat tissues.

**Figure 4 nutrients-15-02490-f004:**
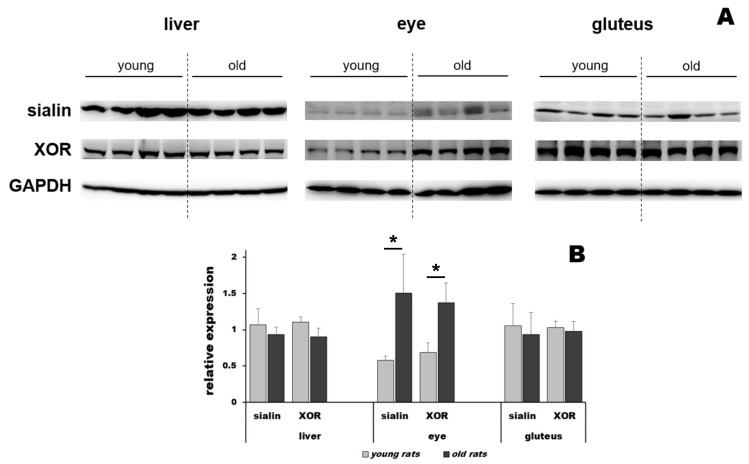
Western blot for sialin and xanthine oxidoreductase (XOR) in the liver, eye and gluteus of young and old rats (**A**); relative expression levels of sialin and XOR in liver, eye and gluteus of young rats (light gray bars) and old rats (dark gray bars) (**B**), n = 4 for old and young rats, ***** denotes *p* < 0.05.

**Figure 5 nutrients-15-02490-f005:**
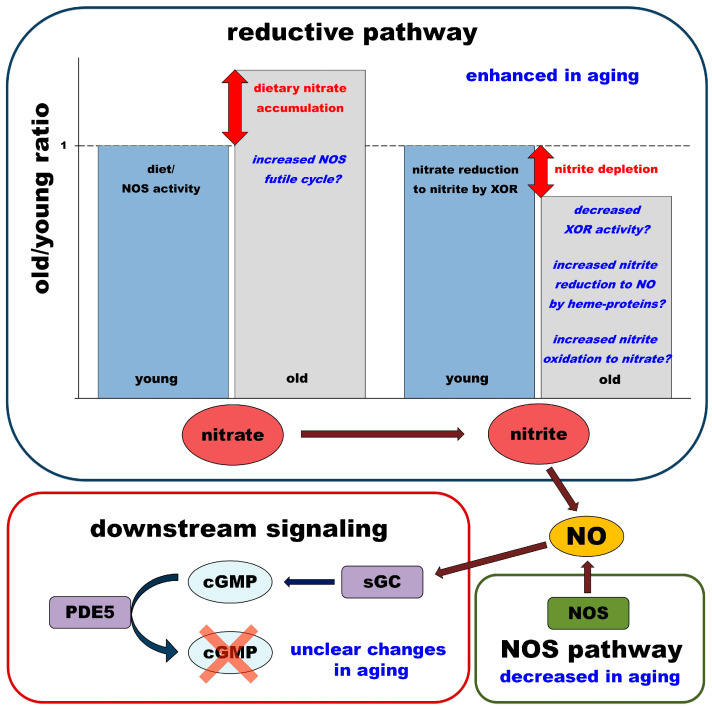
Schematic representation of differences in the NO metabolic pathway of old and young rats at baseline levels. Levels of nitrate and nitrite in young rats are plotted as “standards,” they are used only as a reference for the relative changes observed/proposed in old rats, with no other associated meaning.

## Data Availability

All data and detailed information about used materials are available upon written request to the corresponding author.

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
