# Peer review of "Nitrate and Nitrite Metabolism in Aging Rats: A Comparative Study"

_nutrients, 2023, doi:10.3390/nu15112490_

Round 1
Reviewer 1 Report
The manuscript by Piknova et al. conveyed a comparative study on nitrate and nitrite levels in old vs. young rats. It addressed the significant question of NO metabolism during the aging process from a physiological level and provided multiple valuable hypotheses for future studies on this topic. Overall, I think this work serves the readers well with the purpose of introducing the basic understanding of nitrate and nitrite metabolism during organismal aging and should be recommended for publication. However, the study overall presents to be over-simplifying certain aspects and there are also certain points that requires improvements or clarification that will help to improve the quality of this article:
Major points:
1. Regarding the dietary supplementation of nitrate, the authors should explain: a) The purpose of this treatment: Is it to mimic medical/dietary interventions for nitrate supplementation if it becomes deficient during aging? Or only to test the response of relative enzymes and functions of the reduction pathway upon nitrate intake? For the latter question, nitrate deficient studies should be conducted or cited from previous studies. b) How is the length and concentration of NaNO3 justified based on the questions in a)?
2. For Figure 3, it might also be relevant to compare “old rat nitrate diet/old rat baseline” vs. “young rat nitrate diet/young rat baseline”, which gives a more direct understanding of how much change is induced by nitrate supplementation in old vs. young rats. In this way, the aging-related factors can be more clearly elucidated. The current way the authors showed emphasized more on the aging related difference on absolute values, which is also an important way to visualize the data but lacks the illustration of the differences in the ability to “change” according to the nitrate supplementation in each age group.
3. For Figure 3, it is also worth mentioning in the discussion section if it is physiologically beneficial or not for the old rat to have a nitrate or nitrite level much higher than the young rat after the dietary supplementation (e.g. nitrate levels in lacrimal glands, salivary glands and nitrite level in the heart)
4. For Figure 4, it will be much more helpful for the readers to see a whole picture of NO metabolism if the authors could also show (or cite previous studies) levels of nitric oxide synthases and other enzymes involved in the reduction pathway (e.g. AO, SO, mARC), and even the molybdenum- or heme-containing proteins that further reduce nitrite into NO. The authors should also show (at least for sialin and XOR) the levels before vs. after nitrate dietary supplementation in each age group, as this will add great value to the question conveyed by this study.
5. The authors should provide a more thorough discussion/hypothesis on why in some organs (e.g. lacrimal glands and salivary glands), the nitrate level in old rats were much higher than young rats after nitrate supplementation, yet this ratio is not the same for the nitrite levels of the same organs after nitrate supplementation.
Minor points:
1. Line 151 and Line 158 mentioned “Table 1A”, which is missing from the manuscript. I am assuming the authors were referring to “Supplementary Table 1A”?
2. In Figure 1, the darker bars should be labeled as “young rat nitrate diet” instead of “young rat nitrate”, which is incomplete. Same for Figure 2.
3. The authors used both the spelling of “aging” and “ageing” throughout the text. There should be a uniformed spelling format, for which the editors could provide a standard.
4. The authors should rewrite/rephrase the first few sentences of the Abstract Section, as it is a direct copy of the beginning of the Introduction Section.
5. The texts in the blue bars in Figure 5 are too small to read if printed.
The sentence structure in the manuscript sometimes appears to be odd, meaning it's different from most manuscript, such as starting a sentence by saying "Figure 1 compares..." instead of directly discussing the result and then referring to Figure 1. This does not hurt the over all presentation of the manuscript. It is just an uncommon way to discuss the result compared to most manuscript.
In Line 333, it is better to use "can not" than "can't". Also, if the evidence can not be provided, it is better to have an explanation for why this can not be done, such as "It is beyond the scope of this manuscript." Otherwise the authors could just phrase it like "This hypothesis requires further experimental evidence to be confirmed/explored."
Author Response
Thank you for your comments, please see attached word file with answers.

Reviewer 2 Report
Nitric oxide (NO) is very important and can regulates many physiological processes in the body. This study investigated the balance of storage, transport and reduction of nitrate/nitrite ions affected/disturbed by processes naturally occurring during aging. It was found that differences in nitrate and nitrite content in tissues of old and young rats at baseline level, with nitrate being in general higher and nitrite lower in old rats, but with no difference in levels of nitrate transporting protein and nitrate reductase between old and young rats, with exception of eye. The research results are very interesting, and here are some of my opinions.
(1) Line 83-84. “As expected, dietary nitrate supplementation led to, in most cases, a significant increase of nitrate and nitrite in studied organs and tissues” What’s the value of the significance?
(2) Line 246-248. The second sentence “We also showed that……” please change to a different expression.
(3) Line 258-259. Please give more discussion
(4) Line 318. “We assume that the nitrate reduction into nitrite is more robust in these organs than in the rest of the body but due to technical reasons we do not have any supporting data for this at this point” This explanation feels very far-fetched.
(5) Line 354. “we believe that the organs of old rats are just as able as young rats to transport, store and use dietary nitrate” dietary nitrate?
(6) Line 358-364. Please give more discussion
(7) Figure 5 and related contents should be placed before the conclusion Conclusion
Author Response
Thank you for your comments, please see attached word file.
